# Long-Term Electroclinical and Employment Follow up in Temporal Lobe Epilepsy Surgery. A Cuban Comprehensive Epilepsy Surgery Program

**DOI:** 10.3390/bs8020019

**Published:** 2018-02-01

**Authors:** Lilia Maria Morales Chacón, Ivan Garcia Maeso, Margarita M. Baez Martin, Juan E. Bender del Busto, María Eugenia García Navarro, Nelson Quintanal Cordero, Bárbara Estupiñan Díaz, Lourdes Lorigados Pedre, Ricardo Valdés Yerena, Judith Gonzalez, Randy Garbey Fernandez, Abel Sánchez Coroneux

**Affiliations:** Epilepsy Surgery Program International Center for Neurological Restoration, 25th Ave, No 15805, Havana, Cuba; ivangarciamaeso@gmail.com (I.G.M.); minou@neuro.ciren.cu (M.M.B.M.); jebender@infomed.sld.cu (J.E.B.d.B.); marugeniagarcian@gmail.com (M.E.G.N.); nquintanal@neuro.ciren.cu (N.Q.C.); baby@neuro.ciren.cu (B.E.D.); lourdesl@neuro.ciren.cu (L.L.P.); rvaldes@neuro.ciren.cu (R.V.Y.); judith@neuro.ciren.cu (J.G.); randis0770@gmail.com (R.G.F.); abel@neuro.ciren.cu (A.S.C.)

**Keywords:** temporal lobe epilepsy, epilepsy surgery, long term follow up, Electroencephalogram interictal epileptiform discharge, employment

## Abstract

The purpose of this paper is to present a long- term electroclinical and employment follow up in temporal lobe epilepsy (TLE) patients in a comprehensive epilepsy surgery program. Forty adult patients with pharmacoresistant TLE underwent detailed presurgical evaluation. Electroencephalogram (EEG) and clinical follow up assessment for each patient were carried out. The occurrence of interictal epileptiform activity (IEA) and absolute spike frequency (ASF) were tabulated before and after 1, 6, 12, 24 and 72 months surgical treatment. Employment status pre- to post-surgery at the last evaluated period was also examined. Engel scores follow-up was described as follows: at 12 months 70% (28) class I, 10% (4) class II and 19% (8) class III-IV; at 24 months after surgery 55.2% (21) of the patients were class I, 28.9% (11) class II and 15.1% (6) class III-IV. After one- year follow up 23 (57.7%) patients were seizure and aura-free (Engel class IA). These figures changed to 47.3%, and 48.6% respectively two and five years following surgery whereas 50% maintained this condition in the last follow up period. A decline in the ASF was observed from the first year until the sixth year after surgery in relation to the preoperative EEG. The ASF one year after surgery allowed to distinguish “satisfactory” from “unsatisfactory” seizure relief outcome at the last follow up. An adequate social functioning in terms of education and employment in more than 50% of the patients was also found. Results revealed the feasibility of conducting a successful epilepsy surgery program with favorable long term electroclinical and psychosocial functioning outcomes in a developing country as well.

## 1. Introduction

Temporal lobe epilepsy (TLE) associated with mesial temporal sclerosis is considered the most common and pharmacoresistant type of epilepsy in adults; therefore, 70–80% of epilepsy surgeries are performed in the temporal lobe. That is the reason why our epilepsy surgery program precisely initiated with this epilepsy type. According to statistics available as of 2005 epilepsy prevalence in Cuba is approximately 3. 1/1000 people [1]. Thus, an estimated 80,000 people in the country suffer from epilepsy, about 24,000 are pharmarcoresistant, and approximately 2400 might have surgery indications. However, the success of epilepsy surgery (ES) depends upon the early identification of potential surgical candidates based on the available resources and technologies [2]. Eventually, after temporal lobe resective epilepsy surgery, the patient is 70% likely to be seizure-free, and over 30% to be free of antiepileptic drug (AEDs) within 2 years after surgery [3,4].

Epilepsy surgery still remains the most underutilized of all acceptable medical intervention not only in developed countries but also in developing countries as the causes of surgical failure are not clearly understood, and the reasons for this are creating an enormous treatment gap [5].

On the other hand, there is little literature available on epilepsy surgery in developing nations and the access to and availability of epilepsy management programs are very limited. Thus, there are multiple social, economic, and medical challenges in establishing successful epilepsy surgery programs in low- and middle-income countries, and the issue of developing epilepsy centers in resource-limited areas in a large scale is essential [6,7,8].

Although the result of epilepsy surgery has improved over time [9,10,11,12], the few studies that have assessed chronological changes in surgical outcome in medial TLE have generally reported cross-sectional analyses limited to seizure outcomes in low- and middle-income countries. Additionally, the Engel and ILAE classification systems address only seizure outcome, and do not assess psychosocial, behavioral, cognitive and vocational development; all vital to gauge the utility of epilepsy surgery [13,14]. Even when considering both cross-sectional and longitudinal studies, there is insufficient research directly examining the longer-term educational and vocational outcomes of adult TLE surgery patients.

This article illustrates the results of a long- term follow up in TLE patients operated in the first comprehensive surgery program carried out at the International Center for Neurological Restoration in Havana, Cuba. A longitudinal Electroencephalogram (EEG), and clinical long term follow up for each patient was assessed. Social functioning outcomes are also addressed in this paper.

## 2. Materials and Methods

### 2.1. Patient Population

Epileptic patients with pharmacoresistant focal epilepsy were referred from different regions of the country. Cases were required to be non-responsive to at least 2 appropriate AEDs trials due to inefficacy and intolerance; hence recurrently compromised by seizures. Then, patients were consecutively admitted to the first comprehensive surgery program from May 2012 to September 2015. Only subjects submitted to TLE resection with over one-year follow-up after surgery were included whereas those with prior brain surgery were left out. Lastly, patients were evaluated by an epileptologist (LM) before being operated by an epilepsy surgeon (IG). Family and patient´s informed consent was received in all cases.

### 2.2. Presurgical Evaluation

Each patient underwent noninvasive presurgical evaluation program including: (a) prolonged video-electroencephalography (VEEG) monitoring with scalp electrodes placed according to the international 10–20 system and additional anterior temporal electrodes; (b) Magnetic Resonance Imaging (MRI) scans with a 1.5 T scanner (Siemens Magnetom Symphony) including the following sequences: T1-weighted images with and without gadolinium-DTPA, T2-weighted images, fluid-attenuated inversion recovery images and magnetization-prepared rapid gradient echo sequences. Axial images were obtained with a modified angulation parallel to the temporal lobe long axis to assess the mesiotemporal structures; (c) A comprehensive battery of neuropsychological tests (attention assessment, memory, higher verbal and visual functions); (d) Perimetric evaluation and quadrant visual evoked potential VEPs [15].

Voxel based morphometric MRI post processing comprising volumetric analysis and functional neuroimaging using interictal and ictal brain single photon emission computed tomography and Magnetic Resonance Spectroscopy (MRS) were carried out in only 15% of patients when MRI was normal, and when there was discordance between VEEG and MRI, in accordance with our previously published protocol [16].

In the VEEG, ictal EEG patterns at seizure onset were categorized Type I, Type II A&B, Type III based on established criteria by Ebersole and Pacia [17]. With respect to the resection site ictal and interictal EEG findings were classified concordant (if 75% or more corresponded to the site of seizure origin (based on seizure semiology, MRI abnormality, and/or area resected) or discordant (i.e., any evidence of a wider, even if lateralized, spatial distribution outside the resection area involving more than one seizure. Ictal EEG activity was categorized localized to the presumed lobe of seizure origin, lateralized to the presumed hemisphere of seizure origin and diffuse (uncertain hemispheric origin). The distribution of interictal epileptiform discharges (IEDs) during prolonged video-EEG monitoring was assessed by analyzing 15 minutes interictal EEG samples every 1 h. The data recorded in relation to events identified by button presses or by seizure or spike detection programs was also reviewed. Patients underwent VEEG monitoring for 10.7 ± 3.14 days.

Ictal and interictal EEG was analyzed by a qualified epileptologist involved in the study (LM).

Patient test results were discussed in an epilepsy surgery conference including a multidisciplinary team.

### 2.3. Surgical Procedure and Resection Size

Anteromedial Temporal lobectomy tailored by Electrocorticography (ECoG) recording was carried out by a neurosurgeon (IG). ECoG data acquisition was performed with a Medicid-5 digital EEG system (Neuronic SA, Cuba) made in Cuba, using AD-TECH subdural electrodes (grid and strips).The superior gyrus was spared with the neocortical resection of the middle and inferior temporal gyrus. The extent of anterior temporal neocortical resection was adjusted according to the ECoG findings. The absolute longitude of lateral and mesial resected tissue was measured by the neurosurgeon (IG) using slices of images in T1, T2 and FLAIR six months after surgery. The lateral (neocortical) aspect included the distance (in mm) between the posterior edge of the internal table and the anterior limit of the resected area considering the middle temporal gyrus in the axial slices of MRI whereas the mesial aspect was calculated from axial slices in parallel with the preserved hippocampus. Two tangential lines were drawn: one between the posterior border of the resection and the contralateral hippocampus, and the other between the anterior limit of the preserved hippocampus and the side of resection. The distance between these tangential lines was the mesial longitude.

### 2.4. Tissue Characterization and Histopathological Examination

Resected specimens varied in size depending on the ECoG result. Haematoxylin-eosin and Kluver-Barrera myelin special stain were performed in mesial and neocortical specimens.

Hippocampal sclerosis (HS) was defined by neuronal loss in CA1, CA3 and CA4 regions of the hippocampus. Gliofibrillary acidic protein (GENNOVA, dilution 1/50) was used to qualitatively evaluate the astrogliosis as a consequence of the neuronal loss in the hippocampus and neocortex as well as the baloon cells. Synaptophysin (GENNOVA, ready to use) was performed when immunohistochemical staining was necessary.

The presence of focal cortical dysplasia (FCD) and HS was independently confirmed by two neuropathologists. For mycroscopic diagnosis, and FCD classification, the system proposed by the International League Against epilepsy was used [18]. For Central Nervous System tumor histopathological diagnosis purpose WHO classification was used [19].

### 2.5. Post Operative Follow-Up

Clinical follow up assessment for each patient was carried out six months (*n* = 40); one (*n* = 40), two (*n* = 38), five (*n* = 37) years; and later from six to fourteen years (*n* = 30) (mean 9.7 years) after surgery 42.5% patients were followed up for over ten years and 75% for at least seven years. Overall, the mean follow-up was 8.6 ± 3.9 years. Data were collected prospectively.

Postsurgical seizure outcome assessment was based on the system proposed by Engel. [Engel class I, free of disabling seizures; class IA, seizure-free; class II, rare seizures (fewer than three seizures per year); class III, worthwhile improvement (reduction in seizures of 80% or more); class IV, no benefit] [20]. To illustrate, and for some statistical analysis, class I was classified as “satisfactory” outcome, while classes II, III and IV as “unsatisfactory” seizure relief outcome.

Digital Scalp 30–60 min routine EEG recording was reviewed and interpreted by experienced electroencephalographers (LM) to assess the presence of interictal epileptiform discharges (IEDs) defined as sharp waves, spikes and electrographic seizure. IEDs were characterized as temporal (ipsilateral or contralateral to surgery), bitemporal, extratemporal, and generalized. Scalp EEG analysis was done using bipolar (longitudinal and transverse with temporal chains) and referential montages. The occurrence of IED, and absolute spike frequency (ASF) calculated as IED/min were then tabulated before and after 1, 6, 12, 24 and 72 months surgical treatment.

In order to investigate social functioning, the employment and educational status pre-to post-surgery, in the last follow up was analyzed.

### 2.6. Statistics Analysis

Data were collected from follow-up visits and sequentially entered into the database. These data were summarized with descriptive statistics for each variable comprising means, medians, and standard deviations for continuous variables and frequencies for categorical variables. Normality of the data was tested using Shapiro-Wilk test. Results showed non-normal distribution of some variables. Besides, non-parametric inference was used for comparisons. The Mann-Whitney U test was used to compare nonparametric values. The Friedman ANOVA and sign test were used to compare the electroclinical follow up one, six months; one, two, five years; and from six to fourteen years following surgery. Differences were considered statistically significant at the 5% level.

### 2.7. Ethical Considerations

All the procedures followed the rules of the Declaration of Helsinki of 1975 for human research, and the study was approved by the scientific and ethics committee from the International Center for Neurological Restoration (CIREN37/2012).

## 3. Results

### 3.1. Demographic Profile and Electroclinical Features

Patients undergoing noninvasive presurgical evaluation at the CIREN comprehensive epilepsy surgery program were carefully selected by our interdisciplinary team of epileptologists. During extracranial Video-EEG monitoring a mean of 13.1 ± 9.8 seizures per patient was recorded with a mean Video -EEG monitoring efficiency equal 0.99. In the whole group the first seizure occurred at day 3; and the third at day 5. Awake and sleep seizures indexes were 0.77 and 0.24 respectively. In 70% (28 of 40) of cases the antiepileptic drug regimen was partially reduced during the video-EEG session. Type I Ictal EEG pattern was documented in 60% of cases; and 62.5% had bitemporal with unilateral predominance IED (See Table 1).

#### 3.1.1. Temporal Lobectomy and Complications

The resection amount in the 40 patients submitted to anterior temporal lobectomy was based on a result combination obtained from presurgical evaluation and intraoperative ECoG findings Table 1. There were no significant differences in the resection size between right and left temporal lobectomies. Mann Whitney U test for neocortical *p* = 0.46, for mesial *p* = 0.18. COMPLICATIONS One case experienced dysphasia and right hemiparesis. Aseptic meningitis was treated postoperatively in two patients. Another patient suffered from bacterial meningitis. Quadrantanopsia was not considered as complication. One patient, categorized in Engel Class II, died from an orthopedic surgery four years after surgery.

#### 3.1.2. Pathology

The most common etiology was Focal Cortical Dysplasia (FCD) associated with a principal lesion (FCD type III). 67.5% had FCD type IIIa (cortical lamination abnormalities in the temporal lobe associated with hippocampal sclerosis); 10% FCD type IIIb (cortical lamination abnormalities adjacent to a glial or glioneuronal tumor; one patient with FCD IIIc (cortical lamination abnormalities adjacent to vascular malformation, and six with only HS. Dual pathology was documented to be associated with pylocitic astrocytom and arachnoid cystic in two of the patients.

### 3.2. Electroclinical Follow Up

After one- year follow up, 23 (57.7%) were completely seizure free and aura free (Engel class IA) while two and five years after surgery the percentage changed from 47.3%, to 48.6% respectively. In the last follow up period 50% of the patients maintained this condition. However, there was no significant difference in the number of patient’s seizure and aura free over the long time. Patients kept their AEDs for at least 2 years postsurgery.

Engel scores follow-up was described as follows: at 12 months 70% (28) class I, 10% (4) class II and 19% (8) class III-IV; at 24 months: 55.2% (21) of cases were class I, 28.9%(11) class II and 15.1% (6) class III-IV. Five years after surgery, 54.05% (20) class I, 35.1% (13) class II, 10.8% (4) class III. At the last follow-up period 55.1% (16) class I, 24.1% (7) class II and 20.6% (6) class III. There was a notable difference between clinical evolutions considering all evaluated period (Friedman ANOVA *p* = 0.01286, The percentage of patients in Engel class I decreased two years postsurgery in relation to the previous year (*p* = 0.01 Sign test). Overall, there was no substantial variation for Engel class I within 24 months and the last follow up period (*p* > 0.05 Sign test) (See Figure 1 and Table 2).

#### Interictal Epileptiform Discharges on Post-Operative EEG

There were noteworthy changes in the ASF follow up (Friedman ANOVA *p* = 0.0108. A decline in the ASF was observed one, two and six years after surgery in relation to the preoperative EEG (*p* 0.003 Sign test). However, there were no differences in the ASF between the first year following surgery and the other evaluated periods. See Figure 2. Interestingly the ASF in the EEG recorded one year postsurgery was significantly different in “satisfactory” outcome cases from those with “unsatisfactory” seizure relief outcome *p* = 0.02 at the last clinical follow up Figure 3.

### 3.3. Pre and Post-Surgery Education and Employment Status in TLE Patients

In terms of education and employment status, 41.6% of TLE operated patients who were employed before surgery remained in regular work. 5% of patients moved to supported work and 7.8% began to study after surgery. A total of 27.7% of the patients remained unemployment whereas 13.8% became unemployed post-surgery. More than 70% of patients in both employment and unemployment group were seizure free. As a whole, over 50% of our patients showed an adequate social functioning in terms of education and employment fourteen years after TLE surgery.

## 4. Discussion

This study indicates a steady number of patients as seizure-free in a long-term temporal lobe epilepsy surgery outcome, and highlights the value of longitudinal postoperative EEG in epilepsy surgery follow up. Our results also provide evidence of an adequate social functioning in terms of education and employment in TLE operated patients.

In TLE patients undergoing surgery Engel class I was achieved in 70% at 12 months’ follow-up, 55.2%, 54.05% and 55.1% at 2, 5, and from 6 to14 years respectively. This representsa seizure decrease during the first year after surgery which remained stable from the second year until the last evaluated period. In agreement with our results, other studies have shown seizure free in the range of 60–70% after temporal lobectomy [21,22,23]. The proportion of patients free of seizure and aura over the long time in the current study (57.7%) was similar compared to a recent report that also showed complete seizure freedom in 65% of patients at 1year and in 56.5% at long-term follow-up of ≥5 years after temporal lobectomy [24]. Another author that included temporal and extratemporal resections detailed that >80% of the patients experienced class I outcome at the last four-year follow-up [25].

Our results are in keeping with other series from developing countries such as Argentina which found Engel class I outcome in 68.21% at 12 months’ follow-up [26]. Mikati MA also stated 70% Engel class I, 9% class II, 14% class III, and 7% class IV after resective surgery in 93 adults and children who had undergone epilepsy surgery involving temporal resections in 54% of the cases at the American University of Beirut [27].

It should be observed that our seizure freedom outcome was better than that reported by other developing countries such as india and Uganda. In India, for instance, excellent seizure outcome (seizure-free or having only auras) was achieved in 7/17 patients (41%) [28] and in Uganda’s series 6/10 60% of patients were seizure-free after temporal lobe epilepsy surgery [29]. Remarkably, Mrabet KH, and Campos et al., reported 100% and 88.24% of patients in Engel class I after epilepsy surgery [30,31]. The former included 15 patients with hippocampal sclerosis in an epilepsy surgery program in Tunisia with French collaboration, whereas the latter comprised 17 cases. A lower number of operated epilepsy patients was reported in all these series. Our results are also comparable with those described in a larger series of 87 children with temporal lobe epilepsy with HS and lesion-related epilepsies with a non-invasive protocol [32].

An interesting finding in this research is that patients undergoing temporal resection experienced significantly favorable long-term outcome. Two of the few studies that have reported longitudinal follow-up during 10 years include a retrospective single-center study in 325 patients (adults and children), in which 48% were continuously seizure-free after five years and 41% after 10 years [33]. The second study reported 55% seizure-free (without or with auras) five years following surgery and 49% 10 years later in 615 adults, 497 with temporal lobe resection [34].

Clearly, such comparisons are limited by selection criteria and referral patterns, which are likely to differ from different centers in Latin American countries. In order to standardize these criteria, our cases were discussed in an epilepsy surgery conference.

Another remarkable fact in our patients was that the epilepsy duration continued for more than 10 years (mean 19 years); and that the number of AED treated was higher than five (3–10). Moreover, most patients had bitemporal IED with unilateral predominance in the preoperative EEG. On the other hand, the most common etiology was FCD associated with a principal lesion (FCD type III). Cortical dysplasia has been documented in histologic specimens removed for treatment of drug resistant TLE epilepsy and there is evidence that a standard anterior temporal lobectomy offers a good seizure outcome [35,36,37]. Corresponding results were 64.2% and 56.8% at 1 and ≥5 years, respectively in patients with isolated mesio temporal lesion, and 66.4% and 56.0%, respectively in patients with mesial TLE and additional FCD [38]. In recent years, FCD has been identified as a major cause of pharmacoresistant focal epilepsy in patients undergoing surgical resection [39,40].The rate of seizure free after resection changed from 52 to 68.9% [41,42,43].

Even with this electroclinical profile, our seizure freedom outcome (Engel class I) was equivalent to other series in developed and developing countries, which points to epilepsy surgery as a long term effective treatment for carefully selected patients with pharmacoresistant temporal lobe epilepsy. It is important to highlight that distinction between this study and other series are limited not only because of the use of the classification proposed by the Task Force of the ILAE but also due to the epilepsy type addressed in this research. The current ILAE classification encompasses the FCD Type III that includes FCD associated with other principal lesions based on previous reports of lesions showing dysplastic changes in histology after resection [44].

In relation to preoperative video EEG it is important to notice that in 60% of the cases, type I ictal EEG pattern was recognized at seizure onset. This pattern also anticipated a favorable clinical evolution five years postsurgery; and has been associated with good outcomes in patients with TLE in earlier studies [28,45]. Nowadays, the controversy about the relative contributions of ictal scalp VEEG, routine scalp outpatient interictal EEG, intracranial EEG and MRI for predicting seizure-free outcomes after temporal lobectomy still remains [24].

Regarding post-operative EEG follow-up, a decline in the ASF was observed one, two and six years after surgery in relation to the preoperative EEG. It should be mentioned that quantitative measures of changes have been applied in very few studies; and the spike frequency in postoperative EEGs has been evaluated with controversial results by other authors [46,47,48,49]. Gropel reported that the spike frequency decreased in fourteen patients with mesial TLE-HS at four months, one and two years after surgical treatment increased in one; but it did not change in seven [48].

It ihas been stated that longer duration EEGs or repeated EEGs performed at different time intervals following surgery might have increased the chance of capturing IED, especially if epochs of sleep were involved. An advantage of this study is that we were able to quantify the IED in different follow up periods.

We also found that the ASF in the EEG recorded one year postsurgery was significantly different in “satisfactory” outcome cases from those with “unsatisfactory” seizure relief outcome. As a whole, studies are conflicting as to whether interictal EEG findings predict seizure recurrence [23,46,47,50,51]. Some of them showed a strong predictive value [47,52], while others revealed no predictive value [48,53,54,55]. Conflicting results are feasible as populations examined varied considerably among studies in the majority of series focusing on anterior temporal lobectomy. Also, there is no standard time to carry out a post-operative EEG, and centers that perform it routinely set their own protocols [49,56].

The assessment of surgical outcome in epilepsy beyond seizure control has received little attention in terms of social functioning, and the studies investigating employment and education in adults are scarce. Moreover, the range of postoperative psychosocial adjustment issues has been well documented, particularly within the first 24 months after surgery.

We found an adequate social functioning in terms of education and employment fourteen years after TLE surgery in more than 50% of the patients. In our study, 41.6% of TLE operated patients who were employed before surgery remained in regular work. 5% of patients moved to supported work and 7.8% began to study after surgery.

Although most studies have investigated employment outcomes alongside measures of quality of life, [57,58,59], a minority of them has focused on employment status pre-and post-surgery [60,61,62,63]. Overall, there is a mix of improvements and reductions in occupational status [64,65]. In the majority of studies an improve of vocational outcomes has been suggested [60,66,67,68,69,70]. Other authors as Asztely and colleagues reported a decline in the number of patients employed full-time following surgery [71]. Reid K and cols demonstrated that employment outcomes are directly relevant to patient satisfaction with surgery up to ten years later [70]. In support of the surgical treatment efficacy some studies comparing surgery to ongoing medical management demonstrate a trend towards higher employment postsurgery [62,72,73,74]. One study that used a healthy control demonstrated that up to 10 years post-surgery, even though patients who were seizure free were more likely to be employed than those with recurrent seizures. Overall, the number of patients that were still working was notably less than healthy controls (61% compared to 96%) [75].

It is not surprising, that in adults deemed eligible for epilepsy surgery, employment is a commonly cited reason for electing to undergo surgery. Both patients and their families identify educational and vocational outcomes as important, with expectations of improvements post-surgery [76,77,78,79].

The principal constraint of this study is the low number of patients that precluded the extraction of valuable information about potential prognostic factors in seizure recurrence so that more cases should be recruited to address this issue. Nonetheless, our results revealed a steady number of patients being seizure-free in a long-term temporal lobe epilepsy surgery outcome, and highlights the value of longitudinal postoperative EEG in epilepsy surgery follow up. Equally, results confirm the possibility of conducting a successful epilepsy surgery program with favorable long term electroclinical and psychosocial functioning outcomes in a developing country as well.

## Figures and Tables

**Figure 1 behavsci-08-00019-f001:**
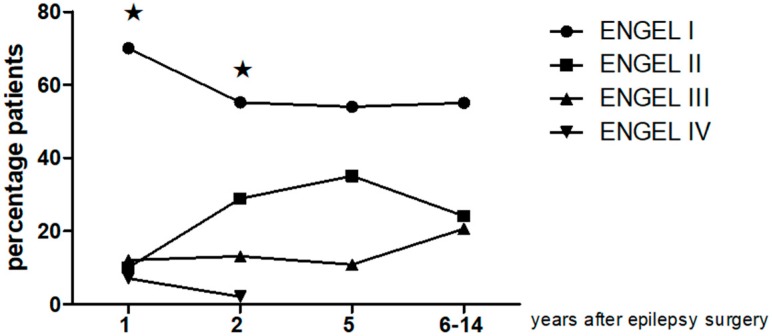
Long term clinical follow up using Engel Scale in temporal lobe epilepsy patients submitted to epilepsy surgery. Notice that the percentage of patients in Engel class I decreased two years postsurgery in relation to the previous year (★ *p* = 0.01, Friedman ANOVA and Sign test). There was no substantial variation for Engel class I within 24 months and the last follow up period (*p* > 0.05).

**Figure 2 behavsci-08-00019-f002:**
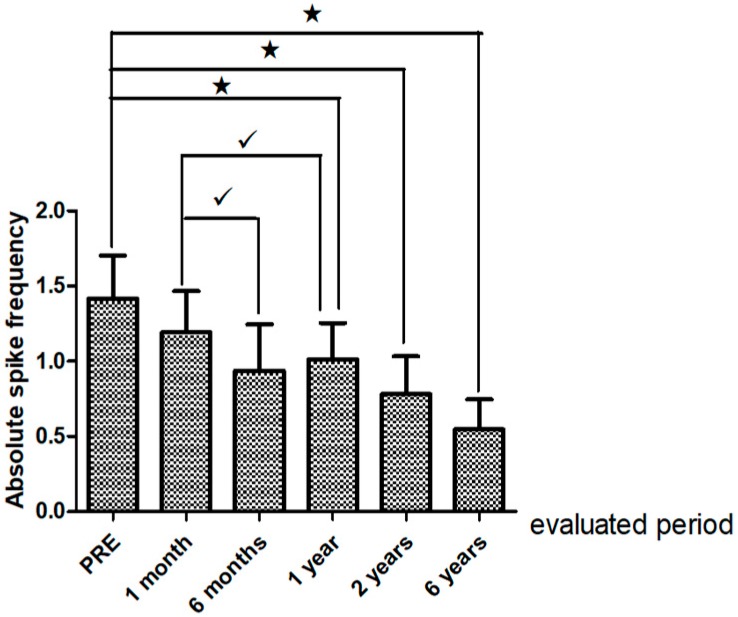
Absolute spike frequency (ASP) on pre-and postoperative Electroencephalogram EEG (one, six months and one, two and six year). Comparisons of pre-EEG with one, two and six years after epilepsy surgery (★ *p* < 0.05). Comparisons between one month postsurgical EEG and six months and one year post surgery (√ *p* < 0.05) Friedman ANOVA and Sigh test. There were no differences in the SAF between the first year after and the other evaluated periods. Notice x axis: evaluated EFG period, y axis: mean and SEM of the ASF (spike/min).

**Figure 3 behavsci-08-00019-f003:**
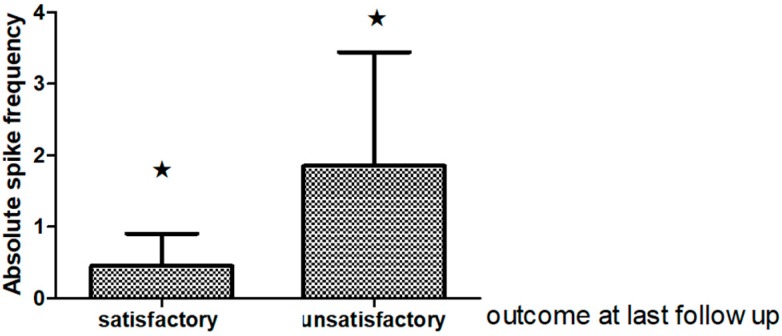
Bar grah showing comparisons of absolute spike frequency on the EEG one year postsurgery (mean and standart deviation SD) in temporal lobe patients with satisfactory (Engel class I) and unsatisfactory (Engel class II–IV) outcome at the last follow up (6–14 years) (★ *p* < 0.05, Mann Whitney U test).

**Table 1 behavsci-08-00019-t001:** Demographic, clinical and surgical cohort characteristics (40 patients).

Mean age at surgery (years ± SD range)	33.5 ± 9.7, (range 16–58)
Mean age at seizure onset (years ± SD range)	13.7 ± 11.3, (range 9–52)
Gender	Male: 22 (55%) Female: 18 (45%)
Mean epilepsy duration (year ± SD range)	19.6 ± 10.18 (range 2–42)
Precipitant event *n* (%)	31 (77.5%), febrile seizures 47.6%
Mean number of antiepileptic drugs tried ± SD (range)	5.95 ± 2.02, (range 3–10)
Resection lateralization *n* (%)	Right 19 (47.5%) Left 21 (52.5%)
Mean follow-up (month range) (year ± SD range)	8.6 ± 3.9 years (range 1–14)
Seizure types	Complex partial seizures: 100%
	Simple partial seizures: 76%
	History of secondary generalized tonic-clonic seizures: 84%
Ictal EEG pattern *n* (%)	Type I (5–9 Hz) 24 (60%)
	Type II (2–5 Hz) 16 (40%)
Ictal EEG topography, *n* (%)	Concordant 31 (77.4%)
	Discordant 8 (22.5%)
Preoperative interictal EEG, *n* (%)	Unilateral/concordant 15 (37.5%); Bilateral with ipsilateral predominance 5:1, 25 (62.5%)
Left hemisphere resection longitude (mean ± SD), Right hemisphere resection longitude (mean ± SD)	mesial 18.23 ± 6.76 mm, neocortical 41.6 ± 9.8 mm
	mesial 15.06 ± 5.19 mm, neocortical 40.19 ± 12.07

SD Standard deviation.

**Table 2 behavsci-08-00019-t002:** Year-by-year Clinical follow up by Engel class.

Follow-Up	Class I Patients *n*, (%)	Class II Patients *n*, (%)	Class III Patients *n*, (%)	Class IV Patients *n*, (%)
1 year, *n* = 40	28, (70%)	4, (10%)	5, (12%)	3 (7%)
2 year, *n* = 38	21, (55.2%)	11, (28.9)	5, (13.1%)	1 (2%)
5 year, *n* = 37	20, (54.05)	13, (35.1%)	4, (10.8%)	
6–14 year mean 9.7y *n* = 29	16, (55.1)	7, (24.1%)	6, (20.6%)

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
