# Peer review of "Long-Term Electroclinical and Employment Follow up in Temporal Lobe Epilepsy Surgery. A Cuban Comprehensive Epilepsy Surgery Program"

_behavsci, 2018, doi:10.3390/bs8020019_

Round 1

Reviewer 1 Report

Behavioral Sciences (behavsci-261676)

Long- term electroclinical and employment follow up in temporal lobe epilepsy surgery. A Cuban comprehensive epilepsy surgery program 

      This manuscript presents a review of temporal lobe resections performed for pharmaco-resistant epilepsy at the International Center for Neurological Restoration in Havana, Cuba, from 2012 to 2015.

      This retrospective study includes details of the pre-surgical evaluation including MRI, neuropsychological testing, video-EEG/EMU (scalp only), and perimetric evaluation with VEPs.  Resection size was studied, as well as histopathology of the resected specimen.  Serial EEGs (scored for absolute spike frequency (ASF)) were performed and follow-up continued for up to 14 years.  Employment history followed as well.

      Forty patients were identified.  These patients’ demographics are described, as well as presurgical, surgical, and post-surgical follow-up as described above.  The principle findings are an Engel Class I outcome in 70% of patients at one year after surgery, and then ~55% for years 2 and beyond.  The ASF is reported prior to the surgery and at multiple time points following surgery.  Employment percentages are reported before and after surgery.

      Overall I feel the manuscript is of interest, though attention to the following points would improve the manuscript.

1.    The organization of the discussion could be greatly improved.  The key findings of the manuscript are not presented clearly, and the comparison to other published series is jumbled.  The conclusions are similarly not clearly stated.  Also, the grammar in the discussion could be improved.

2.    Line 229.  While there is a seizure decrease over the first two years after resection, this result should be put into context of the decreasing percentage of patients with an Engel Class I outcome in Year 1 as compared to Year 2.

3.    The comparisons for ASF are unclear, both in the text and in the figure.  Also, a difference in the ASF in those with “satisfactory” vs “unsatisfactory” outcomes is reported, but no data is presented in the text or figures.

4.    In Figure 2 some of the comparisons which are reported as significant (e.g. PRE vs 2 years) appear to have nearly the same mean and standard deviation.

5.    Comparing correlations between “satisfactory” vs “unsatisfactory” outcomes and

6.    Overall there are many typographical and spelling errors, some of which are listed below.

Minor issues

1.    Please be consistent with either comma or period as the decimal mark.

2.    Line 155, perhaps W test or W-test

3.    Line 168.  Spelling error, extracranial.

4.    Line 173.  It is not clear to what “ipsilateral” refers; I assume ipsilateral to the eventual resection.  Perhaps unilateral would be more clear.

5.    Table 1, row 2.  Please add units to range for age of onset.

6.    Table 1, row 5.  Add space between event and n.

7.    Table 1, row 6.  Add SD for anti-seizure drugs tried.

8.    Line 190. Spelling errors.

9.    Figure 2.  No units on y-axis of ASF graph.

10. Line 257. Typographical error.

Author Response

Review 2

Line number refers to the first manuscript version submitted in some lines there are number modification in the current manuscript . I enclosed the new  manuscript version 

The organization of the discussion was improved.  The key findings of the manuscript and conclusions were modified.

Line 229 The percentage of patients in Engel class I decreased two years postsurgery in relation to the previous  year  ( p=0.01 Sign test ). Overall, there was no substantial variation for Engel class I within 24 months and the last follow up period ( p>0.05 Sign test )

The comparisons for ASF were explained in detail. The text and Figure 2 was modified. A decline in the ASF was observed one, two and six  years after surgery in relation to the preoperative EEG (p=0.003 Sign test).  However, there were  no differences in the ASF between the first year following surgery and  the  other evaluated periods. See Figure 2. 

The difference in the ASF in those with “satisfactory” vs “unsatisfactory” outcomes was  presented in the text and a Figure 3 was added.

Reviewer 2 Report

The authors report on the long-term post operative follow up in TLE patients operated at the International Center for Neurological Restoration in Havana, Cuba. The followup stretches up to 14 years (in a small subset of patients), the results are interesting and worth reporting. however I have some major concerns that hinder my enthusiasm. 

My first major concern is that the manuscript is hard to follow and in some instances key information is lacking. I will list the major points below, but I feel that the whole manuscript needs major revision:

- line 38 "According to statistics available as of 2005 epilepsy prevalence in Cuba is approximately 3, 1-7, 5/1000 people", what do the numbers "3" and "1-7" mean?

- line 54: "Although the result of epilepsy surgery has improved over time [9-12], chronological changes in surgical outcome in medial TLE in low- and middle-income countries are also restricted" It is not clear to me what is restricted.

- line 78: "monitoring with scalp electrodes placed according to the international 10–20 system and additional scalp electrodes" what are the additional scalp electrodes for?

- line 87: "In some cases, if the MRI was normal" I think the authors should be more precise -in this instance as well as in other cases throughout the manuscript- and switch from general indications (i.e. some cases) to a quantitative statement (i.e. report the percentage).

- line 99: "The distribution of interictal epileptiform discharges (IEDs) during prolonged video-EEG monitoring 98 was assessed by 15-second interictal EEG samples visual analysis every 15 min" It is not clear to me how the sampling worked, did the authors visually inspected only 15 seconds every 15 minutes of recordings? If so, why such a small portion of the recordings and on the base of which criteria were the analyzed segments selected?

- line 124: how did the authors estimate the neuronal loss? was this conclusion based on the Haematoxylin–eosin staining? if so, how did they quantify the neuronal loss?

- line 199: "The first seizure during examination occurred at day 3; and the third seizure at the fifth day." unclear, does this sentence refer to one patient or to all patients? 

- Table 1, fourth entry "mean epilepsy duration": 19, 6±10, 18 (range 2-42), no clear what the number "19" is; is the mean duration 6 years and SD 10 years?

- Table 1, sixth entry "Mean number of antiepileptic drugs tried", the SD is missing.

- Line 179: "There were no significant differences between right and left temporal lobectomies" what is the aspect hat does not differ between left and right lobectomies? their number?

- Line 192 and 197: I am confused, if Engel class I means seizure free (as per methods section), what is the percentage of seizure free patients? 57% (line 192) or 70% (line 197).

- Figure 1: add asterisk to denote significance at year 2;

My second major concern is about Figure 2, the authors report significant change in aboslute spike frequency between PRE and post operative year 1, 2, and 6. However the variation at 2 and 6 years seems high and I am puzzled by the significance level reached. Has he p value been corrected for multiple comparison? how many patients were analysed at every time point? Furthermore, the y axes does not report units which makes harder to interpret the data.

Author Response

Answers

Review 1

Line number refers to the first manuscript version submitted in some lines there are number modification in the current manuscript . I enclosed the new  manuscript version 

Line 38 According to statistics available as of 2005 epilepsy prevalence in Cuba is approximately  3. 1 and 7. 5/1000 people [1].

Line 78 .Each patient underwent noninvasive presurgical evaluation program including: (a) prolonged video-electroencephalography (VEEG) monitoring with scalp electrodes placed according to the international 10–20 system and additional anterior temporal electrodes.

Line 87 Voxel based morphometric MRI post processing comprising volumetry analysis and functional neuroimaging using interictal and ictal brain single photon emission computed tomography and Magnetic Resonance Spectroscopy (MRS) were carried out in  only 15% of patients when  MRI was normal, and when there was discordance between VEEG and MRI,  in accordance with  our previously published  protocol

Line 99 The distribution of interictal epileptiform discharges (IEDs) during prolonged video-EEG monitoring was assessed by analyzing  15 minutes  interictal EEG samples every 1 hour  . The data recorded in relation to events identified by button presses or by seizure or spike detection programs was also reviewed.  Patients underwent VEEG monitoring for 10,7 ±3,14 days.

Line 124 Hippocampal sclerosis (HS) was defined by neuronal loss in CA1, CA3 and CA4 regions of the hippocampus. Gliofibrillary acidic protein (GENNOVA, dilution 1/50) was used to qualitatively evaluate the astrogliosis as a consequence of the neuronal loss in the hippocampus and neocortex as well as the baloon cells.  Synaptophysin (GENNOVA, ready to use) was performed when immunohistochemical staining was necessary.

Line 199 . In the whole group the first seizure occurred at day 3; and the third  at day 5. Awake and sleep seizures indexes were 0.77 and 0.24 respectively. In 70 % (28 of 40) of cases the antiepileptic drug regimen was partially reduced during the video-EEG session .

Line 179 There were no significant differences in the resection size between right and left temporal lobectomies. Mann Whitney U test for neocortical p =0, 46, for mesial p =0. 18.

Line 192 After one- year follow up, 23 (57.7%) were completely seizure free and aura free (Engel class IA) while two and five years after surgery the percentage changed from 47. 3%, to 48. 6% respectively. In the last follow up period 50% of the patients maintained this condition

Line 197 Engel scores follow-up was described as follows: at 12 months 70% (28) class I, 10% (4) class II and 19% (8) class III-IV; at 24 months: 55.2% (21) of cases were class I, 28.9 %( 11) class II and 15.1% (6) class III-IV. Five years after surgery, 54. 05 %( 20) class I, 35. 1% (13) class II, 10., 8% (4) class III. At the last follow-up period 55. 1 % (16) class I, 24. 1% (7) class II and 20. 6 % (6) class III.

Figure 1 and 2 were modified. A Figure 3 was added

Round 2

Reviewer 2 Report

The manuscript quality improved significantly. As a minor comment the reference list is not numbered.